# Assessment of Genetic Variability and Bran Oil Characters of New Developed Restorer Lines of Rice (*Oryza sativa* L.)

**DOI:** 10.3390/genes13030509

**Published:** 2022-03-13

**Authors:** Mamdouh M. A. Awad-Allah, Azza H. Mohamed, Mohamed A. El-Bana, Samira A. F. El-Okkiah, Mohamed F. M. Abdelkader, Mohamed H. Mahmoud, Mohamed Z. El-Diasty, Manal M. Said, Sahar A. M. Shamseldin, Mohamed A. Abdein

**Affiliations:** 1Agricultural Research Center, Rice Research Department, Field Crops Research Institute, Giza 12619, Egypt; momduhm@yahoo.com; 2Department of Agricultural Chemistry, Faculty of Agriculture, Mansoura University, Mansoura 35516, Egypt; azza@ufl.edu; 3Citrus Research & Education Center, University of Florida, IFAS, Lake Alfred, FL 33850, USA; 4Agricultural Research Center, Food Technology Institute, Giza 12619, Egypt; m.elbana733@yahooo.com; 5Department of Agriculture Botany, Faculty of Agriculture, Kafr El-Sheikh University, Kafr El-Sheikh 33516, Egypt; samira.fouad@agr.kfs.edu.eg; 6Department of Plant Production, College of Food and Agriculture, King Saud University, Riyadh 12372, Saudi Arabia; mohabdelkader@ksu.edu.sa; 7Department of Biochemistry, College of Science, King Saud University, Riyadh 11451, Saudi Arabia; mmahmoud2@ksu.edu.sa; 8Genetic Department, Faculty of Agriculture, Mansoura University, Mansoura 35516, Egypt; m_z_diasty@mans.edu.eg; 9College of Biotechnology, Misr University for Science and Technology, Giza 12563, Egypt; manalfaris@yahoo.com; 10Botany Department, Women’s College for Arts, Science and Education, Ain Shams University, Cairo 11566, Egypt; shams.sahar@women.asu.edu.eg; 11Biology Department, Faculty of Arts and Science, Northern Border University, Rafha 91911, Saudi Arabia

**Keywords:** rice, rice bran oil, genetic variability, coefficient of variation, genetic advance

## Abstract

Rice is one of the most important crops in Egypt. Due to the gap between the demand and the availability of the local edible oils, there is need to raise the nutritional value of rice and, therefore, to improve the nutritional value of the consumer. This research was carried out at the Experimental Farm of Sakha Agricultural Research Station, Sakha, Kafr El-Sheikh, Egypt, during the 2019 and 2020 seasons. Five newly developed genotypes of rice, namely NRL 63, NRL 64, NRL 65, NRL 66, and Giza 178 as check variety (control), were used to evaluate the analytical characterization of raw rice bran and rice bran oil from rice bran, study the genetic variability and genetic advance for various quantitative and qualitative traits in rice as well as, rice bran oil. The genotypes were evaluated in a randomized complete block design (RCBD) with three replications. Analysis of variance revealed highly significant variations among the genotypes for all the studied characters. Data revealed that high estimates of the phenotypic coefficient of variance (PCV%) and genotypic coefficient of variance (GCV%) were observed for amylose content percentage, peroxide value (meq/kg oil), myristic C14:0, and arachidic C20:0, indicating that they all interacted with the environment to some extent. The line NRL66 and NRL64 showed the highest and high values of mean performance for grain yield (t/h), grain type (shape), amylose content percentage, crude protein, ether extract and ash of milled rice, crude protein, ether extract, ash, phosphorus, magnesium, manganese, zinc, and iron of stabilized rice bran oil. Genetic advance as a percentage of mean was high for most of the studied traits. It indicates that most likely, the heritability is due to additive gene effects, and selection may be effective. The percentage of advantage over the Giza 178 as the commercial variety was significant and highly significant among the genotypes for all the characters studied in the two years, indicating that the selection is effective in the genetic improvements for these traits.

## 1. Introduction

Rice (*Oryza sativa* L.) is the second main cereal crop and staple nourishment food of half of the people in terms of global production 740.96 million tons of rough rice in 2014, which provides approximately 70 MMT of bran. In Egypt, rice is the second food crop, and the production of rice in Egypt was 6.00 million tons [1,2,3]. Rice bran is a by-product produced during the rice milling process and accounts for 5–10% of the milled rice [4]. It is a suitable source of protein (14–16%), fat (12–23%), crude fiber (8–10%), carbohydrates, vitamins, minerals, essential unsaturated fatty acids, and phenolics [5,6,7]. Because of its nutritional superiority, abundant micronutrients, longer shelf life, as well as being stable at higher temperatures and giving better flavor to foodstuffs, rice bran oil is usually used as an excellent cooking medium. The amount of rice bran components differs as a function of rice type, climatic conditions, storage conditions, rice bran stabilization, and processing methods [8]. In addition, it typically contains 88–89% neutral lipids, 3–4% waxes, 2–4% free fatty acids, and approximately 4% unsaponifiable [9]. However, the use of rice bran oil is limited due to its enzymatic activity after rice de-hulling. Rice bran is rich in lipids, and intense lipase activity in the presence of endogenous lipoxygenase causes rapid deterioration of these lipids by rancification [10]. Because of lipid susceptibility, the commercial use of rice bran requires enzymatic inactivation immediately after bran separation to avoid fatty acid liberation, extend its shelf life and allow its commercialization for human consumption [11,12].

The fatty acid profile of rice bran oil reveals about 19% saturated (palmitic acid), 42% monounsaturated (oleic acid), and 39% polyunsaturated (linoleic acid), so rice bran oil is one of the healthiest and most nutritious edible oils [13].

The percentage of oil in rice bran is from 18% to 23%. It is an oil rich in essential fatty acids, and it is rich in nutritional components such as dietary fire, vitamin B and E, and minerals such as iron, calcium, potassium, chlorine, magnesium, and manganese [14].

From a nutritional point of view, the interest in rice bran oil has been growing, mainly because of its health benefits, which include a reduction in both serum and LDL cholesterols [15,16]. The healthy vegetable oil of rice bran oil is a suitable source of various antioxidants such as oryzanol, tocopherols, tocotrienols, squalene, and phytosterols. This healthy vegetable oil also has greater oxidative stability and longer shelf life than other vegetable oils. This healthy oil is also a rich source of monounsaturated fatty acids (n-9 MUFA), n-6 PUFA, and sterols, as well as it has been shown to reduce bad cholesterol.

This healthy vegetable oil with a balanced fatty acid profile is more effective for preventing heart disease, skin disease, and cancer, and it improves the immune system activity and neurological function [17,18]. The characterization of genetically broad rice germplasms for both bran lipid content and fatty acids composition is of special importance in identifying possible sources of variation as well as potentially beneficial genotypes.

Plant breeding is the continuous endeavor to develop superior plant phenotypes that are better adapted to human needs by using the available genetic variation. Plant breeding aims to improve the quality, diversity, performance of food, industrial, and other economically important crops. Rapid advances using conventional breeding techniques led to Green Revolution, when a remarkable increase in the production of rice by the development of high-yielding varieties [19].

Genetic variability, which is due to the genetic differences among individuals within a population, is the main aim of plant breeding programs because proper management of diversity can produce a permanent gain in the performance of the plant and can buffer against seasonal fluctuations [20]. Genetic variability among traits is important for breeding and in selecting desirable types. As the breeders are interested in a selection of superior lines based on phenotypic performance, the foremost function of heritability is its predictive role in representing the reliability of phenotypic performance as an indicator of breeding value and in providing information on the transmission of character from the parent to progeny. Heritability studies provide opportunities for breeders to predict the interaction of genes in successive generations and are essential for effective breeding programs.

For effective genetic improvement of grain yield, it is important to understand how the proportion of genetic components and genetic advances are affected by environments. Thus, genetic advance is yet another important selection parameter that aids breeders in a selection program. Estimates of genetic advances will help in knowing the nature of gene action affecting the concerned traits [21].

The expected production of rice bran oil in Egypt can be worked out to be 150 thousand tons produced from 6 million tons of rice annually. This will enhance decreasing the gap between demand and availability of local edible oils [1]. So, the present investigation was designed to assess the analytical characterization of crude rice bran and rice bran oil from rice bran of newly developed genotypes, namely NRL63, NRL64, NRL65, NRL66, and Giza 178 as check variety.

## 2. Materials and Methods

### 2.1. Plant Materials

Five genotypes of rice include four newly developed restorer lines, namely NRL 63, NRL 64, NRL 65, NRL 66, and Giza 178 as check variety (control). These lines were selected from a set of 122 iso-cytoplasmic restorer lines of rice were developed from two promising rice hybrids, IR79156A/86945-L (3 lines), followed by G46A/Giza 178 (1 line). The selection procedure was started in 2015 as F_2_ up to F_7_ in 2020, where 4 staple lines were selected according to bran oil content. Further experiments were conducted to estimate all characters of bran oil for the developed lines as well as Giza 178 as a local check. The genotypes names of the materials studied are presented in Table 1.

During the 2019 and 2020 growing seasons, the five genotypes were grown in a randomized complete block design (RCBD) with three replicates. Each replication had 5 rows. The length of the row was 5 m in length and 20 cm between rows; each row had 25 individual plants.

Data were collected on grain yield per hectare (tons), grain shape (L/B ratio), amylose content%, chemical composition (%) of milled rice (moisture, crude protein, ether extract, ash, available carbohydrates), gross chemical composition (%) of stabilized rice bran (moisture, crude protein, ether extract, ash, crude fiber, available carbohydrates), contents of (Ca, Mg, Fe, Cu, Zn, Na, K, and Mn), some physical and chemical properties of crude rice bran oil (refractive index (25 °C), specific gravity (25 °C), acid value (%), peroxide value (meq/kg oil), iodin value (gI/100 g oil), saponification value (mg KOH/g oil), unsaponifiable matter (%)), fatty acids composition of rice bran oil (myristic C14:0, palmitic C16:0, palmitoleic C16:1, stearic C18:0, oleic C18:1, linoleic C18:2, linolenic C18:3, arachidic C20:0, eicosenoic C20:1, TSFA%, and TUSFA%).

### 2.2. Grain Shape

Rice grain is classified into three grain (kernel) shapes: short, medium, and long grain. Classification of grain shapes is based on length to width ratios of rice kernels. The length to width ratios of the paddy (rough rice) of the long, medium, and short-grain rice are measured [22]:**Grain Type Length/Width Ratio**Long ≥ 3.4Medium ≥ 2.3Short ≤ 2.2

### 2.3. Amylose Content

The amylose content of the native rice starch was determined according to the method described for the analysis of milled rice amylose content by the authors of [23].

### 2.4. Processing of Rice Bran

Different substances such as husk may be present in the bran. Hence, the full fatted raw bran was sieved, which removes husk. The samples thus obtained were free from impurities.

### 2.5. Stabilization of Rice Bran

Rice bran samples were stabilized by autoclave under atmospheric pressure for 10 min at 120 °C according to the method described by the authors of [24]. Finally, bran samples were stored in dark conditions at −10 °C in water insusceptible containers until further analyses.

### 2.6. Rice Bran Analysis

Rice bran oil was extracted according to the method described by the authors of [25].

#### 2.6.1. Determination of Gross Chemical Composition

Moisture, ether extract, crude protein (*N* × 5.95), ash, and crude fiber contents were performed according to the Association of Official Analytical Chemists [26]. Available carbohydrates were determined by difference according to the methods of [26]. Minerals contents (Ca, Mg, Fe, Cu, Zn, and Mn) were determined according to the methods outlined in the work of [26] using atomic absorption spectrophotometer (Perkin Elmer Model 4100 ZL), Perkin Elmer Inc., Wellesley, MA, USA, while (Na and K) were determined using a flame photometer, London, UK. On the other hand, phosphorus was determined by the ascorbic acid technique using the colorimetric method.

#### 2.6.2. Determination of Fatty Acids Composition of Rice Bran Oil Samples

The methyl esters were prepared using benzene: methanol: concentrated sulfuric acid (10:86:4), and the methylation process was carried out for one hour at 80–90 °C according to the work of [27]. Identification of the fatty acid methyl esters was performed by gas-liquid chromatography (G.L.C A) Pye Unicam gas-liquid chromatography (model PU4550), “(Diagramma AG, Dietikon, Switzerland)” equipped with a flame ionization detector and coiled glass column (1.6 m × 4 mm) packed with 10% PEGA (polyethylene glycol adipate) supported on chromosorb W-AW 100–200 mesh. Samples (1–1.5 uL) into the column using ahamilton microsyringe. Gas chromatographic conditions used for isothermal analysis were column 190 °C flow rates: hydrogen 33 mL/min, nitrogen 30 mL/min, and air 330 mL/min. Peak areas were measured using spectto physic integrator [26].

### 2.7. Statistical Analysis

#### 2.7.1. ANOVA Test

The Data Were Statistically Analyzed Using the Analysis of Variance (ANOVA) a Model Proposed by the Authors of [28].

The magnitude of the components of variances has been obtained from the analysis of variance to appraise the different genetic parameters as described by the works of [29,30]. The genotypic and phenotypic variances were calculated as per the formulas proposed by the authors of [31]. The genotypic (GCV%) and phenotypic (PCV%) coefficient of variation was calculated by the formulas given by the authors of [31]. Heritability in a broad sense [h^2^(_bs_)] was calculated by the formula given by the authors of [32] as suggested by the authors of [33]. From the heritability estimates, the genetic advance (GA) was estimated by the following formula given by the authors of [33].

Mean squares were used to estimate:σ^2^_g_ = (MSS − MSE)/r 
where: MSS: mean sum of squares due to treatments, MSE: mean sum of squares due to error from the analysis of variance, and r: number of replications:σ^2^_ph_ = σ^2^_e_ + σ^2^_g_
where broad-sense heritability (h^2^_bs_) was estimated as follows:h^2^_bs_ = (σ^2^_g_/σ^2^_ph_) × 100
and the phenotypic (PCV) and genotypic (GCV) coefficients of variation were computed as follows:PCV = 100 × √σ^2^_ph_/X^−^
GCV = 100 × √σ^2^_g_/X^−^
GA = k × h^2^_bs_ × √σ^2^_ph_

Expected genetic advance (GA): expected genetic advance from direct selection for all studied traits was calculated according to the work of [29] as follows:GA% at 5% (selection intensity) = 100 × k × h^2^_bs_ × σ^2^_ph_/X^−^
or:GA% = (GA/X^−^) × 100
where X^−^: general mean and k is selection differential (k = 2.06 for 5% selection).

Ref. [34] categorized the value of GCV and PCV as: low = 0–10%; moderate = 10–20%; and high = >20%.

As suggested by the authors of [33], h^2^_bs_ estimates were categorized as low = 0–30%; medium = 30–60%; and high = above 60%.

#### 2.7.2. The Advantage over Commercial Variety

The advantage over the high-yielding commercial variety calculated as percentage of increased or decreased of the newly restorer lines over the commercial one (*CK*).

The advantage over commercial variety (ACK) = $\frac{\bar{M}-CK}{CK}$ × 100.

Appropriate LSD values were calculated to test the significance of the advantage over the commercial variety, according to the method:$$L.S.D\;\mathrm{for}\;(\mathrm{ACK})=t_{0.01}^{0.05}\;\sqrt{\frac{2\mathrm{MSe}}{r}}$$
where:*t*: Value at certain probability level and given degrees of freedom for error.MSe: Error mean squares from the analysis of variance.r: Number of replications.$\bar{M}$: The mean of the newly developed restorer lines for a character.

This method is described by the authors of [35].

## 3. Results

### 3.1. Mean Performance

The mean performances for rice grain yield, grain shape (L/B ratio), and amylose content percentage traits of the studied genotypes during the 2018 and 2019 growing season are presented in Table 2.

Regarding the grain yield (ton/h), the results revealed that the genotypes NRL 63, NRL 66, NRL 65, and NRL 64 showed the highest mean values 14.13, 13.51, 12.00, 12.04 and 13.18, 12.83, 12, 11.25, 11.03, in the second and first season, respectively for grain yield ton/h. While the check rice variety Giza 178 provided the lowest mean values of the grain yield ton/h, its values were 9.74 and 9.63 ton/h, in the second and first season, respectively.

The lines NRL 66, NRL 63, NRL 65, NRL 64, and Giza 178 (check variety) showed desirable mean values toward medium paddy grain shape with an average of 2.63, 2.7, 2.72, 2.74, and 2.97 over two seasons, respectively, Table 2.

Regarding the amylose content (%), NRL 66 recorded the desirable low mean values in two years, 22.72 and 21.83. While the check rice variety Giza 178 provided the lowest mean value 18.46 and 17.4) in the two years, respectively. The obtained results indicated that the line NRL 66 was the best for grain quality traits, Table 2.

### 3.2. Chemical Composition of Some Rice Genotypes

Data presented in Table 3 showed that the moisture content of milled rice ranged from 11.31% to 12.67% in the first year. While in the second year, the moisture content of milled rice ranged between 11.19% and 12.56%. From the same table, it could be observed that NRL 64 had the highest level of crude protein, 8.23% and 8.13%, in the first and the second years, respectively. In contrast, the lowest values were recorded in milled rice of Giza 178 variety 7.47% and 7.43% in the two years, respectively.

Results of the same table also revealed that there was a significant difference in ether extract between the different genotypes. Milled rice of NRL 63 had the highest ether extract content 0.99% and 0.97% in comparing with the other tested samples at the two years. In contrast, milled rice of Giza 178 variety had the lowest level of ether extract content 0.7% and 0.66% at the two years, respectively. High differences in ash content were recorded between the genotypes. Moreover, milled rice of NRL 64 had the highest level of ash content of 0.98 in the first year, respectively. The data presented in the same table showed that the milled rice had the highest carbohydrates content in variety Giza 178 compared with the other tested samples.

### 3.3. Proximate Chemical Composition of Stabilized Genotypes Rice Bran (g/100 g on a Dry Weight Basis)

The chemical composition of stabilized bran of rice genotypes was determined, and the results are tabulated in Table 4. The obtained results indicate highly variation in the moisture content of rice bran samples among the selected rice genotypes. Stabilized NRL 65, rice bran has the highest moisture content, 8.87% and 8.82%, at the two years, respectively. It could be seen from Table 4 that the crude protein of stabilized rice bran, NRL 66, contained the highest content of crude protein, which was 17.85% and 17.75%, followed by stabilized rice bran of NRL 63, which was recorded at 17.38% and 17.30%, while the lowest value of crud protein 16.36% and 16.30% for stabilized rice bran was observed in NRL 64 at the two years, respectively. Results also from the same table showed that the ether extract content ranged from 21.46% to 23.22% and 21.46% to 23.22% at the two years, respectively. The stabilized rice bran of NRL 66 and NRL 63 have higher levels of crude oil content than those of stabilized rice bran NRL 64 and NRL 65. Data in Table 4 showed that stabilized rice bran genotypes contain 8.8% to 9.25% and 8.76% to 9.21% ash content, and 35.49% to 37.71% and 35.45% to 37.66% available carbohydrate, at the two years, respectively. In addition, stabilized rice bran NRL 66 has the highest crude fiber content, 15.37% and 15.33%, in the two years, respectively.

### 3.4. Mineral’s Content (mg/100 g) of Genotypes Rice Bran

Rice bran is a suitable source of minerals, Table 5, which are present in varying amounts. The major minerals in the genotypes of rice bran were potassium and phosphorous. Potassium content ranged from 787 to 921 and from 782 to 910 mg/100 g in the first and the second year, respectively. In contrast, phosphorous ranged from 860 to 1000 and from 850 to 990 mg/100 g in the first and the second year, respectively. Compared to the selected genotypes and the control, stabilized rice bran NRL 66 had the highest amount of potassium 910 and 901 mg/100 g in the first and the second year, respectively. While the check variety Giza 178 showed the highest values, 921 and 910, at the first and second years, respectively. Results also from the same table showed that the levels of magnesium in the bran ranged from 122.17 to 147.29 mg/100 g in the first year.

The highest magnesium content was observed in NRL 66 rice bran 147.29 mg/100 g in the first year. The levels of calcium in the bran ranged from 29.27 to 38.31 mg/100 g rice bran sample at the first year. The highest calcium content was observed in rice bran of NRL 63, 38.31 mg/100 g, in the first year. Furthermore, the iron levels ranged between 6.86 and 8.61 mg/100 g rice bran samples in the first year. The line NRL 66 showed higher iron content of rice bran compared to the other rice lines in this study. Apparent also from the same table that stabilized rice bran of NRL 66 had the highest elements content in comparison with the other tested genotypes. Concerning the levels of sodium in the bran ranged from 5.61 to 7.70 mg/100 g rice bran sample at the first year. The highest sodium content was observed in rice bran of NRL 63 and NRL 66, with the values 7.70 and 7.63 mg/100 g, in the first year. Furthermore, the manganese levels varied within a range of 4.51–5.45 mg/100 g rice bran samples at the first year. The line NRL 64 showed higher manganese content of rice bran compared to the other rice lines in this study. Regarding the levels of zinc ranged from 3.21 to 4.25 and 3.01 to 3.61 mg/100 g samples at the first and second years, respectively. The highest zinc content was observed in rice bran of NRL 64 and NRL 65, with the values 4.25 and 4.04 mg/100 g, in the first year.

The levels of copper ranged from 0.72 to 0.98 and 0.60 to 0.81 mg/100 g samples at the first and second years, respectively. The highest copper content was observed in rice bran of NRL 66 and NRL 65, with the values 0.98 and 0.92 mg/100 g, in the first year. The same trend of results appeared in the second year.

### 3.5. Some Physicochemical Characteristics of Oils Extracted from Genotypes Rice Bran

The crude rice bran oil of the selected genotypes was compared with the crude oil of Giza 178 variety and analyzed for various physicochemical parameters. The data presented in Table 6 indicated that the physicochemical characteristics of crude rice bran oil for genotypes varied in the middling range. The data in Table 6 showed that the refractive index of rice bran oil samples genotypes ranged from 1.4478 to 1.4693 in the first year and 1.4468 to 1.4689 in the second year. On the other hand, the refractive index of the crude oil of Giza 178 variety was 1.4588 and 1.4587 in the first and second years, respectively. The specific gravity of rice bran oil samples genotypes ranged from 0.9144 to 0.9296 and ranged from 0.9142 to 0.9292 at the first and second years, respectively. In contrast, the specific gravity of the crude oil of the Giza 178 variety was 0.9155 and 0.9151 at the first and second years, respectively. The acid value of crude oil extracted from genotype rice bran samples was different and ranged from 2.245 to 2.889 and 2.221 to 2.872 mg of KOH/g of oil, at the first and second years, respectively. Meanwhile, the acid values of Giza 178 crude rice bran oil were lower, 1.927 and 1.91 mg of KOH/g in the first and second years, respectively. Peroxide values of crude oil extracted from genotype rice bran samples were different and ranged from 1.541 to 1.961 and from 1.533 to 1.951 meq/kg, at the first and second years, respectively. Furthermore, the results indicated that the peroxide value of Giza 178 crude rice bran oil was lower at 1.108 and 1.100 meq/kg, at the first and second years, respectively. It is evident from the results in Table 6 that the iodine value of crude rice bran oil samples of genotype was different and ranged from 106.62 to 114.24 and 106.60 to 114.22 (g/100 g) at the first and second years, respectively. The iodine value of Giza 178 crude rice bran oil was 109.22 and 109.20 g/100 g, at the first and second years, respectively. The obtained results in Table 6 indicated that the saponification value of crude rice bran oil samples genotype was ranged from 183.63 to 187.90 and 183.12 to 187.42 mg KOH/g, at the first and second years, respectively. While the saponification value of Giza 178 crude rice bran oil was lower 181.17 and 180.80 mg KOH/g, at the first and second years, respectively. The same Table 6 showed that unsaponifiable matter of genotypes rice bran oils were 3.43% to 3.93% and 3.31% to 3.81% at the first and second years, respectively.

### 3.6. Fatty Acids Composition (Weight%) of Rice Bran Oil

The data of the fatty acids composition in the rice bran oil showed that oleic acid (C18:1), linoleic acid (C18:2), and palmitic acid (C16:0) are dominant fatty acids in stabilized rice bran oil. The values were in a range of (40.94 and 42.88), (35.167–36.120), and (19.04–20.04), respectively, in the first year (Table 7). While, in the second year, oleic acid (C18:1), linoleic acid (C18:2), and palmitic acid (C16:0) were ranged 39.91 to 41.81, 32.781 to 34.922, and 18.30 to 19.24, respectively.

Data in Table 7 showed that stabilized rice bran oil NRL 66 had the highest palmitic, stearic, and oleic acid content at the first and second years, respectively, in comparing with the other tested genotypes of rice bran.

Total saturated fatty acids percentage (TSFA%) in rice bran oil of NRL 63, NRL 64, NRL 65, and NRL 66 were 22.95, 23.16, 23.26, and 25.28, respectively, in the first year. On the other hand, the total saturated fatty acids percentage (TSFA%) in rice bran oil of NRL 63, NRL 64, NRL 65, and NRL 66, in the second year were 21.51, 21.61, 21.7, and 23.52, respectively, while the total unsaturated fatty acids percentage (TUSFA%) in rice bran oil of NRL 66, NRL 65, NRL 64, and NRL 63 were 79.7, 80.55, 80.67, and 80.78, respectively at the first year. On the contrary, the total unsaturated fatty acids percentage (TUSFA%) in rice bran oil of NRL 66, NRL 65, NRL 64, and NRL 63 were 77.48, 78.3, 78.41, and 78.47, respectively at the second year. The data in Table 7 showed that, for myristic acid (C14:0), NRL 66 provided the highest values 0.63 and 0.66, palmitoleic acid (C16:1), NRL 65 provided the highest values 0.52 and 0.59, arachidic acid (C20:0), NRL 65 provided the highest values 0.960 and 0.961, eicosenoic acid (C20:1), NRL 63 provided highest values 1.097 and 0.992, at the first and second year, respectively. While, for linolenic acid (C18:3), Giza 178 provided the highest values 2.2 and 2.213, followed by NRL 63 showed the highest values 2.18 and 2.19, at the first and second year, respectively.

### 3.7. Analysis of Variance

The analysis of variance is shown in Appendix A. The results obtained that highly significant differences among the genotypes for all the characters studied except refractive index (25 °C) in the two years and the total saturated fatty acids percentage (TSFA%) in the second year showed significant differences among the genotypes, genotypes, indicating that there is variability among the studied lines and would respond positively to selection. The presence of genetic variability is a prime requirement in any crop improvement program. Moreover, the CV of most of the studied traits showed the highest values.

### 3.8. Phenotypic, Genotypic Coefficient of Variation and Genetic Advance

The phenotypic coefficient of variation was higher than the genotypic coefficient of variation for all studied traits. A close examination of experimental results revealed a high estimate of phenotypic and genotypic coefficient of variation for amylose content (%), peroxide value (meq/kg oil), myristic C14:0, and arachidic C20:0, Table 8, Table 9, Table 10 and Table 11.

A moderate value of the phenotypic and genotypic coefficient of variation was observed for grain yield t/h, ether extract and ash of milled rice, calcium, sodium, zinc, iron, and copper of stabilized rice bran oil, acid value (%), palmitoleic C16:1, stearic C18:0, and linolenic C18:3.

However, the low values of the phenotypic and genotypic coefficient of variation were observed for grain shape, gross chemical composition (%), and some minerals content (mg/100 g) of stabilized rice bran. In addition, some physical, chemical properties, and fatty acids profiles showed the lower value of the phenotypic and genotypic coefficient of variation, Table 8, Table 9, Table 10 and Table 11.

High estimates of genetic advance were observed for grain yield (t/h), amylose content (%), moisture, crude protein, and available carbohydrates of milled rice, moisture, crude protein, ether extract, crude fiber, and available carbohydrates of stabilized rice bran oil, phosphorus, potassium, magnesium, calcium, sodium, iron, zinc and manganese of stabilized rice bran oil, acid value (%), peroxide value (meq/kg oil), iodine value (gI/100 g oil), saponification value (mg KOH/g oil), palmitic C16:0, oleic C18:1, linoleic C18:2, linolenic C18:3, TSFA%, eicosenoic C20:1 and TUSFA%, Table 8, Table 9, Table 10 and Table 11. Moreover, moderate genetic advances were observed for grain shape, unsaponifiable matter (%), stearic C18:0, and arachidic C20:0 of stabilized rice bran oil, Table 8 and Table 11. While, the low genetic advances were observed for ether extract and ash of milled rice, ash and copper content of stabilized rice bran oil, refractive index (25 °C), specific gravity (25 °C), myristic C14:0, palmitoleic C16:1, and picosenoic C20:1 Table 9, Table 10 and Table 11.

The data in Table 8, Table 9, Table 10 and Table 11 showed that the genetic advance in percentage (expected) of mean was high for grain yield (t/h), amylose content (%), ether extract, and ash of milled rice, sodium, zinc, iron, and copper of stabilized rice bran oil, acid value (%), peroxide value (meq/kg oil) and myristic C14:0, palmitoleic C16:1, stearic C18:0, linolenic C18:3, arachidic C20:0 of stabilized rice bran oil. Moreover, moderate genetic advances were observed for grain shape, moisture, phosphorus, potassium, magnesium, calcium, manganese of stabilized rice bran oil. While low genetic advances were observed for moisture, crude protein and available carbohydrates of milled rice, crude protein, ether extract, ash, crude fiber, and available carbohydrates and stabilized rice bran oil, refractive index (25 °C), specific gravity (25 °C), iodine value (gI/100 g oil), saponification value (mg KOH/g oil), unsaponifiable matter (%), palmitic C16:0, oleic C18:1, linoleic C18:2, eicosenoic C20:1, TSFA%, and TUSFA%.

### 3.9. The Advantage over Giza 178 Commercial Variety

The data showed that the percentage of advantage over the Giza 178 commercial variety was significant and highly significant among the genotypes for all the studied characters in the two years of the study. These results indicated that the selection is effective in the genetic improvements for these traits (Table 12, Table 13, Table 14 and Table 15). The lines NRL 63, NRL 66, NRL 64, and NRL 65 showed a highly significant percentage advantage over Giza 178 commercial variety for grain yield, with values of 45.2%, 38.7%, 23.6%, and 23.2%, respectively, in the second year, Table 12.

Regarding the crude protein of milled rice, the lines NRL 65 and NRL showed a highly significant percentage advantage over the Giza 178 commercial variety, with values 7.1 and 10.2 in the first year, as well as the percentage 6.7% and 9.4% in the second year. The obtained results indicated that lines NRL 63, NRL 64, NRL 65, and NRL 66 showed highly significant and significant values for ether extract of milled rice in the first and in the second year with values 47%, 31.8%, 19.7%, and 7.6% in the second year.

Moreover, ether extract of stabilized rice bran in the studied lines showed an advantage over Giza 178 commercial variety from 5.1% to 8.2% in NRL 66 and NRL 64 at the first and second year, respectively. On the contrary, NRL 64 32.4% and 20% and NRL 65 25.9% and 16.6% showed the highest values of the percentage of advantage over Giza 178 commercial variety for zinc of stabilized rice bran at the first and second year, respectively. While NRL 66 obtained 23.4% and 23.7% and NRL 65 11% and 10.9% showed the highest values of the percentage of advantage over Giza 178 commercial variety for iron of stabilized rice bran in the first and in the second year, respectively.

## 4. Discussion

### 4.1. Mean Performance

The studied lines showed better performance over the check rice variety for grain yield, grain shape (L/B ratio), and amylose content percentage traits. So, these restorer lines can be used as a source for developing new hybrid combinations and varieties in rice breeding programs. The authors of [34] reported that the selection of parents is a crucial step in breeding programs for improving new lines. Therefore, we can use this genotype to improve some new hybrids for suitable grain quality traits with high yielding. The mean performance of the rice genotypes indicated that NRL 66 was promising concerning the yield performance associated.

#### 4.1.1. Chemical Composition of Some Rice Genotypes

The values of moisture content of milled rice are in line with those of [36]. Some studied lines showed high crude protein content in milled rice; rice grains of these lines are considered as a suitable source for protein in humans’ nutritional diet. This suggested that the selection for this trait could be an effect for improvements of protein content. Ash content has an important role in the determination of the mineral content of rice [37]. High differences in ash content and ether extract content were recorded between the genotypes. So, these lines can be used as a source for developing new hybrids and lines with more than ash content and ether extract content in rice breeding programs. The obtained results were in line with those reported by the authors of [36].

#### 4.1.2. Proximate Chemical Composition of Stabilized Genotypes Rice Bran (g/100 g on a Dry Weight Basis)

Many factors are affecting the chemical composition of rice bran, such as the variety of rice, variation in organic compounds of the soil, fertilizers applied, climatic and environmental factors in addition, degree of milling, and the used treatments [12]. The data in this study showed high differences between the studied lines for moisture content in the selected genotypes, rice bran samples. These obtained results follow those from the work of [38,39]. Furthermore, moisture content plays a greater role during storage [40].

The crude protein of stabilized rice bran showed high differences between studied lines. Whereat, NRL 66 contains the highest content of crude protein, which was (17.85% and 17.75%) followed by stabilized rice bran of NRL 63, which recorded (17.38% and 17.30%) while the lowest value of crude protein (16.36% and 16.30%) for stabilized rice bran was observed in NRL 64 at the two years, respectively. These results are in line with those of [41,42]. For the ether extract content, studied lines have higher levels of crude oil content than those of stabilized rice bran Giz178, which means an improvement in this trait. Significant varietals effect on ether extract content of rice bran as observed here is also reported by the authors of [16] they investigated the crude oil content of 204 rice varieties. They mentioned that the genotype and environment (year) significantly affected oil content, which extended from 17.0% to 27.5%. Stabilized rice bran genotypes showed a wide range of ash content and available carbohydrates at the two years. In addition, stabilized rice bran NRL 66 has the highest crude fiber content (15.37% and 15.33%) for the two years, respectively. The present findings are found to be like the reports of the works of [39,41,43,44].

#### 4.1.3. Minerals Content (mg/100 g) of Genotypes Rice Bran

The mineral composition of rice grain depends considerably on the availability of soil nutrients during crop growth and is generally present in higher levels in the bran layer of rice kernel [44,45]. Rice bran is a suitable source of minerals [6,7]. The most important objectives of this study were to improve the grain oil content, rice bran oil content, grain quality, and high yield potential to improve the nutrition content of consumer rice grains. The present findings are found to be similar to the reports [24,43].

#### 4.1.4. Some Physicochemical Characteristics of Oils Extracted from Genotypes Rice Bran

Refractive index is one of the important physical parameters used in the identification of fat and oils; it could be used for estimation of the degree of saturation of oils. The results are agreed with that obtained by the authors of [46]. The studied lines showed the highest values more than the check variety. The iodine value indicates the stability of oil toward oxidation. It is observed that the higher the iodine value, the greater the degree of unsaturation. Generally, either the unsaturation degree of the fatty acid chains increase or decrease in chain length of the fatty acids tend to increase the specific gravity [47]. The acid value reflects the degree of oil hydrolysis and the amount of free fatty acids (FFAs) in the sample. Higher values indicate undesirable changes as it not only results in greater refining losses but also increases the susceptibility of soils to rancidity. The peroxide values of crude rice bran oil samples are close to the recommended value since it has been reported that peroxide values of freshly extracted oils should be below 10 meq/kg and that the taste of rancid oil appeared clearly when peroxide values were between 20 and 40 meq/kg [26]. A similar trend has been registered by the authors of [48]. They reported a range from 3.0 to 4.5 meq/kg for crude rice bran oil was extracted by n-hexane and by supercritical CO_2_ extraction; however, lower peroxide values ranged from 1.50 to 3.00 meq/kg of crude rice bran oil was observed by the authors of [49]. The iodine value also indicates the stability of oil toward oxidation. It is observed that the higher the iodine value, the greater the degree of unsaturation. These results are comparable with these reported by the authors of [39,49]. Saponification value reflects the average molecular weight of the fatty acids existing in oil. The oil with a low average molecular weight of fatty acids has a higher saponification value. The results of saponification value are in line with those obtained by the authors of [12]. These values were slightly lower than those found by the authors of [50], they found the saponification value of crude rice bran oil was 193.54 mg KOH/g. In addition, unsaponifiable matter, including hydrocarbons, sterols, vitamins, and pigments, usually play an important role in oil stability. The results of unsaponifiable matter were following those reported by the authors of [12,18]. These values were lower than those found by the authors of [49], who reported that the range of unsaponifiable matter of four varieties of rice bran oil was 4.98–6.15%.

#### 4.1.5. Fatty Acids Composition (Weight%) of Rice Bran Oil

Fatty acids are the integral constituents of every fat or oil. The degree of complexity of the glycerides depends on the number of fatty acids and their amounts, and the chemical behavior of lipids largely depends upon their fatty acid constituents. The concentration of major fatty acids C18:1, C18:2, and C16:0 of the investigated rice bran oils generally agreed with those obtained by the authors of [16]. They studied the fatty acid composition of 204 rice varieties and found that the main fatty acids in rice bran oil were palmitic, oleic, and linoleic acids, which were in the ranges of 13.9–49.2%, 22.1–35.9%, and 27.3–41.0%, respectively. The fatty acid profile of rice bran oil is nearly comparable to that of peanut oil but slightly higher in saturation level than that of soybean oil. Therefore, rice bran oil is closely suitable for general frying and cooking applications [51]. The high amounts of unsaturated fatty acids, especially essential fatty acids, lead to an increase in the nutritional values of rice bran oil. The results of the fatty acids composition of rice bran oil extracted from stabilized genotypes rice bran of the present study agreed with those obtained by the authors of [18,39,52].

### 4.2. Analysis of Variance

The highly significant differences among the genotypes were observed for all the studied characters. This suggested that there is an inherent genetic difference among the genotypes, indicating that there is variability among the studied lines and would respond positively to selection. The presence of genetic variability is a prime requirement in any crop improvement program. The set of genotypes used in the present study indicated the existence of significant differences among them for all the studied characters. These results agreed with the results reported by the authors of [53,54,55]. In addition, the CV of the studied traits indicates the existence of a high variation for most studied traits.

### 4.3. Phenotypic, Genotypic Coefficient of Variation and Genetic Advance

The estimates of means, phenotypic coefficient of variation (PCV), genotypic coefficient of variation (GCV), and genetic advance as revealed from results indicated the existence of a considerable amount of variability among the genotypes for all the characters studied. The expected amount of genetic advance can be estimated by the genotypic coefficient of variation along with heritability, as suggested by the authors of [31]. The genotype coefficient was always lower than the phenotype variance of the different traits on the congruence of all the traits under this study. In this study, the phenotypic coefficient of variation was higher than the genotypic coefficient of variation for all studied traits, indicating the influences of environmental factors on these traits. A similar observation has also been noted by the authors of [54,56,57].

A close examination of experimental results revealed a high estimate of phenotypic and genotypic coefficient of variation for amylose content (%), peroxide value (meq/kg oil), myristic C14:0, and arachidic C20:0, indicating that they all interacted with the environment to some extent. Indicated that most likely, the heritability is due to additive gene effects, and selection may be effective.

A moderate value of the phenotypic and genotypic coefficient of variation was observed for grain yield t/h, ether extract and ash of milled rice, calcium, sodium, zinc, iron, and copper of stabilized rice bran oil, acid value (%), palmitoleic C16:1, stearic C18:0, and linolenic C18:3, indicating that they all moderately interacted with the environment to some extent, at an average rate.

While the traits recorded lower value of the phenotypic and genotypic coefficient of variation such as grain shape, this finding is expected due to the concentration of breeder selection for the short-grain, i.e., selection to a limited class, which leads to less variation. In addition to increasing the degree of genetic relationship between the lines under study, [54,57] it has been reported that high genetic advance and genotypic coefficient of variation were observed for most of these characters.

In general, the phenotypic coefficient of variation was higher than the genotypic coefficient of variation, suggesting an influence of environment on the expression of these characters. However, a narrow magnitude of difference between phenotypic and genotypic coefficients of variation for all studied characters suggested a limited role of environmental variation in the expression of these characters. Therefore, selection based on the genotypic performance of the characters would be effective in bringing about considerable improvement in these characters. Selection in the breeding programs based on measurements of phenotypic traits and genotypic variability is measured through analysis of variance [56,58].

Genetic advance gives information on the improvement required in the genotypic value of the new population over the original population. High PCV, high GCV values, and high genetic advance were recorded for these traits suggesting further improvement of lines for these characters for further selection and subsequent use in a breeding program.

The data showed that the genetic advance in percentage (expected) of mean was high for grain yield (t/h), amylose content (%), ether extract, and ash of milled rice, sodium, zinc, iron, and copper of stabilized rice bran oil, acid value (%), peroxide value (meq/kg oil) and myristic C14:0, palmitoleic C16:1, stearic C18:0, linolenic C18:3, arachidic C20:0 of stabilized rice bran oil. It indicates that most likely, the heritability is due to additive gene effects, and selection may be effective. Moreover, moderate genetic advances were observed for moisture, phosphorus, potassium, magnesium, calcium, manganese, stearic C18:0 of stabilized rice bran oil. However, low genetic advances were observed for grain shape, moisture, crude protein and available carbohydrates of milled rice, crude protein, ether extract ash, crude fiber, and available carbohydrates and stabilized rice bran oil, refractive index (25 °C), specific gravity (25 °C), iodine value (gI/100 g oil), saponification value (mg KOH/g oil), unsaponifiable matter (%), palmitic C16:0, oleic C18:1, linoleic C18:2, eicosenoic C20:1, TSFA%, and TUSFA%. Similar results were also reported by the authors of [53,54,56,57,59,60,61].

### 4.4. The Advantage over Giza 178 Commercial Variety

The data in Table 12, Table 13, Table 14 and Table 15 showed that the percentage of advantage over Giza 178 commercial variety was significant and highly significant among the genotypes for all the studied characters in the two years. This proves that the selection is effective in the genetic improvements for these traits. It would be useful to use these newly developed restore lines as a source for developing new promising hybrids and lines in rice breeding programs for these traits.

## 5. Conclusions

In conclusion, analysis of variance showed that there are significant differences among the genotypes for all the characters under study. This indicated that there is scope for the selection of promising genotypes from the present set of genotypes for yield and other traits improvement. The genetic advance in the percentage of mean was high for grain yield (t/h), amylose content (%), ether extract and ash of milled rice, sodium, zinc, iron, copper, acid value (%), and peroxide value (meq/kg oil) of stabilized rice bran oil. Moreover, the genetic advance in the percentage of mean was moderate for grain shape, phosphorus, potassium, magnesium, calcium, magnesium of stabilized rice bran oil. The new lines showed a significant and highly significant percentage of advantage over Giza 178 commercial variety for most of the studied characters in the two years, indicating that the selection is effective in the genetic improvements for these traits. The lines NRL 66 and NRL 64 showed an advantage over Giza 178 commercial variety from 5.1% to 8.2% for ether extract of stabilized rice bran. On the other hand, the lines NRL 63 and NRL 66 showed advantage values 36.9% and 33.2% for grain yield (t/h) in the first and in the second year, respectively.

## Figures and Tables

**Table 1 genes-13-00509-t001:** Names and parentage of the genotypes studied.

Name	Parentage
NRL 63	IR79156A/86945-L
NRL 64	IR79156A/86945-L
NRL 65	IR79156A/86945-L
NRL 66	G46A/Giza 178
Giza 178 (local check)	Giza175/Milyang 49

**Table 2 genes-13-00509-t002:** Mean performances for rice grain yield, grain shape (L/B ratio), and amylose content percentage traits of the studied genotypes during the 2019 and 2020 growing seasons.

Genotypes	NRL 63		NRL 64		NRL 65		NRL 66		Giza 178	
Traits	2019	2020	2019	2020	2019	2020	2019	2020	2019	2020
Grain yield (ton/h)	13.18 ^c^	14.13 ^c^	11.03 ^ab^	12.04 ^b^	11.25 ^b^	12 ^b^	12.83 ^c^	13.51 ^c^	9.63 ^a^	9.74 ^a^
Paddy grain shape	2.67 ^a^	2.73 ^b^	2.71 ^a^	2.77 ^b^	2.7 ^a^	2.74 ^b^	2.67 ^a^	2.59 ^a^	2.9 ^b^	3.03 ^c^
A.C%	29.3 ^c^	28.5 ^c^	29.8 ^c^	29.13 ^c^	30.8 ^c^	29.77 ^c^	22.72 ^b^	21.83 ^b^	18.46 ^a^	17.41 ^a^

Different letters in the same row indicate that the data are significantly different at *p* < 0.05.

**Table 3 genes-13-00509-t003:** Mean performances for chemical composition (%) of milled rice grains of the studied genotypes during the 2019 and 2020 growing seasons.

Parameters	NRL 63		NRL 64		NRL 65		NRL 66		Giza 178	
	2019	2020	2019	2020	2019	2020	2019	2020	2019	2020
Moisture	12.06 ^c^	11.95 ^b^	11.71 ^ab^	11.63 ^b^	12.49 ^c^	12.35 ^c^	11.31 ^a^	11.19 ^a^	12.67 ^c^	12.56 ^c^
Crude Protein	7.65 ^a^	7.55 ^a^	8.23 ^c^	8.13 ^c^	8.00 ^bc^	7.93 ^bc^	7.77 ^ab^	7.70 ^ab^	7.47 ^a^	7.43 ^a^
Ether Extract	0.9933 ^c^	0.7033 ^d^	0.9367 ^c^	0.880 ^cd^	0.8367 ^ab^	0.7967 ^c^	0.810 ^ab^	0.7567 ^ab^	0.7033 ^a^	0.660 ^a^
Ash	0.93 ^b^	0.74 ^b^	0.98 ^b^	0.79 ^b^	0.84 ^a^	0.65 ^a^	0.80 ^a^	0.61 ^a^	0.81 ^a^	0.62 ^a^
Total carbohydrate	90.81 ^ab^	90.74 ^b^	90.24 ^a^	90.21 ^a^	90.70 ^ab^	90.63 ^ab^	91.08 ^bc^	90.98 ^b^	91.68 ^c^	91.62 ^c^

Different small letters in the same row indicate that the data are significantly different at *p* < 0.05.

**Table 4 genes-13-00509-t004:** Mean performances for gross chemical composition (%) of stabilized rice bran samples of the studied genotypes during the 2019 and 2020 growing seasons.

Chemical Composition	NRL 63		NRL 64		NRL 65		NRL 66		Giza 178	
	2019	2020	2019	2020	2019	2020	2019	2020	2019	2020
Moisture	7.31 ^a^	7.28 ^a^	8.47 ^c^	8.42 ^b^	8.87 ^c^	8.82 ^b^	7.75 ^b^	7.70 ^a^	8.48 ^c^	8.40 ^b^
Crude Protein	17.38 ^c^	17.30 ^c^	16.36 ^a^	16.30 ^a^	16.56 ^a^	16.51 ^a^	17.85 ^d^	17.75 ^d^	16.95 ^b^	16.90 ^b^
Ether Extract	22.79 ^b^	22.75 ^bc^	23.22 ^c^	23.15 ^d^	23.07 ^c^	23.02 ^cd^	22.54 ^b^	22.49 ^b^	21.46 ^a^	21.40 ^a^
Ash	9.25 ^c^	9.21 ^c^	8.95 ^ab^	8.91 ^ab^	8.90 ^ab^	8.83 ^ab^	9.02 ^b^	8.98 ^b^	8.80 ^b^	8.76 ^a^
Crude fiber	15.12 ^bc^	15.11 ^b^	14.50 ^a^	14.47 ^a^	14.95 ^b^	14.90 ^b^	15.37 ^c^	15.33 ^b^	15.33 ^c^	15.28 ^b^
Available Carbohydrates	35.71 ^a^	35.63 ^a^	37.21 ^bc^	37.17 ^bc^	36.77 ^b^	36.74 ^b^	35.49 ^a^	35.45 ^a^	37.71 ^c^	37.66 ^c^

Different small letters in the same row indicate that the data are significantly different at *p* < 0.05.

**Table 5 genes-13-00509-t005:** Mean performances for minerals content (mg/100 g) of stabilized rice bran samples of the studied genotypes during the 2019 and 2020 growing seasons.

Minerals	NRL 63		NRL 64		NRL 65		NRL 66		Giza 178	
	2019	2020	2019	2020	2019	2020	2019	2020	2019	2020
Phosphorus (P)	1000 ^d^	990 ^d^	960 ^c^	950 ^c^	900 ^b^	890 ^b^	860 ^a^	850 ^a^	990 ^d^	980 ^d^
Potassium (K)	806 ^b^	800 ^b^	859 ^c^	851 ^c^	787 ^a^	782 ^a^	910 ^d^	901 ^d^	921 ^d^	910 ^d^
Magnesium (Mg)	136.22 ^b^	134.20 ^b^	122.17 ^a^	120.15 ^a^	143.24 ^c^	141.22 ^c^	147.29 ^d^	145.27 ^d^	134.19 ^b^	132.17 ^b^
Calcium (Ca)	38.31 ^c^	36.13 ^c^	29.27 ^a^	27.07 ^a^	33.25 ^b^	31.10 ^b^	35.22 ^b^	33.17 ^b^	34.22 ^b^	32.07 ^b^
Sodium (Na)	7.70 ^c^	6.83 ^b^	6.30 ^a^	5.61 ^a^	7.21 ^b^	7.11 ^b^	7.63 ^c^	7.51 ^c^	6.50 ^a^	5.70 ^a^
Manganese (Mn)	4.79 ^a^	4.51 ^a^	5.45 ^c^	5.17 ^c^	4.90 ^a^	4.61 ^a^	5.23 ^b^	4.92 ^b^	5.34 ^bc^	5.01 ^c^
Zinc (Zn)	3.42 ^b^	3.11 ^ab^	4.25 ^e^	3.61 ^c^	4.04 ^d^	3.51 ^c^	3.63 ^c^	3.23 ^b^	3.21 ^a^	3.01 ^a^
Iron (Fe)	7.09 ^a^	6.93 ^a^	6.86 ^a^	6.65 ^a^	7.75 ^b^	7.55 ^b^	8.61 ^c^	8.42 ^c^	6.98 ^a^	6.81 ^a^
Copper (Cu)	0.72 ^a^	0.60 ^a^	0.84 ^b^	0.71 ^b^	0.92 ^bc^	0.76 ^bc^	0.98 ^c^	0.81 ^c^	0.86 ^b^	0.72 ^b^

Different small letters in the same row indicate that the data are significantly different at *p* < 0.05.

**Table 6 genes-13-00509-t006:** Mean performances for some physical and chemical properties of crude rice bran oil from some rice genotypes (dry weight basis) of the studied genotypes during the 2019 and 2020 growing seasons.

Physical and Chemical Properties	NRL 63		NRL 64		NRL 65		NRL 66		Giza 178	
	2019	2020	2019	2020	2019	2020	2019	2020	2019	2020
Refractive index (25 °C)	1.4560 ^ab^	1.4540 ^ab^	1.4662 ^b^	1.4663 ^b^	1.4478 ^a^	1.4468 ^a^	1.4693 ^b^	1.4689 ^b^	1.4588 ^ab^	1.4587 ^ab^
Specific gravity (25 °C)	0.9144 ^a^	0.9142 ^a^	0.9165 ^b^	0.9161 ^a^	0.9254 ^c^	0.9251 ^b^	0.9296 ^d^	0.9292 ^c^	0.9155 ^ab^	0.9151 ^a^
Acid value (%)	2.428 ^c^	2.411 ^c^	2.245 ^b^	2.221 ^b^	2.639 ^d^	2.622 ^d^	2.889 ^e^	2.872 ^e^	1.927 ^a^	1.910 ^a^
Peroxide value (meq/kg oil)	1.671 ^c^	1.660 ^c^	1.821 ^d^	1.821 ^d^	1.541 ^b^	1.533 ^b^	1.961 ^e^	1.951 ^e^	1.108 ^a^	1.100 ^a^
Iodin value (gI/100 g oil)	110.33 ^c^	110.31 ^c^	111.18 ^d^	111.17 ^d^	114.24 ^e^	114.22 ^e^	106.62 ^a^	106.60 ^a^	109.22 ^b^	109.20 ^b^
Saponification value (mg KOH/g oil)	187.23 ^cd^	186.71 ^cd^	186.33 ^c^	185.80 ^c^	183.63 ^b^	183.12 ^b^	187.90 ^d^	187.42 ^d^	181.17 ^a^	180.80 ^a^
Unsaponifiable matter (%)	3.43 ^a^	3.31 ^a^	3.87 ^bc^	3.75 ^b^	3.93 ^c^	3.81 ^b^	3.63 ^ab^	3.52 ^a^	3.58 ^a^	3.46 ^a^

Different small letters in the same row indicate that the data are significantly different at *p* < 0.05.

**Table 7 genes-13-00509-t007:** Mean performances for fatty acids profile of stabilized rice bran oil of the studied genotypes during the 2019 and 2020 growing seasons.

FattyAcids	NRL 63		NRL 64		NRL 65		NRL 66		Giza 178	
	2019	2020	2019	2020	2019	2020	2019	2020	2019	2020
Myristic C14:0	0.4700 ^a^	0.4167 ^a^	0.5600 ^b^	0.5067 ^b^	0.670 ^cd^	0.6100 ^c^	0.7133 ^d^	0.6567 ^d^	0.6333 ^c^	0.6067 ^c^
Palmitic C16:0	20.04 ^c^	19.26 ^c^	19.60 ^c^	18.80 ^b^	19.08 ^a^	18.47 ^a^	20.02 ^c^	19.28 ^c^	20.07 ^c^	19.29 ^c^
Palmitoleic C16:1	0.55 ^c^	0.53 ^c^	0.41 ^a^	0.39 ^a^	0.61 ^d^	0.59 ^d^	0.57 ^c^	0.55 ^cd^	0.48 ^b^	0.46 ^cd^
Stearic C18:0	2.07 ^b^	1.95 ^b^	1.77 ^a^	1.63 ^a^	2.01 ^b^	1.863 ^b^	2.26 ^c^	2.157 ^c^	1.96 ^b^	1.840 ^b^
Oleic C18:1	40.94 ^a^	39.91 ^a^	42.00 ^b^	40.90 ^b^	42.12 ^b^	41.15 ^bc^	42.88 ^b^	41.83 ^c^	42.26 ^b^	41.17 ^bc^
Linoleic C18:2	36.120 ^c^	34.922 ^c^	35.49 ^bc^	34.29 ^bc^	35.17 ^bc^	34.013 ^abc^	33.88 ^a^	32.781 ^a^	34.583 ^ab^	33.451 ^ab^
Linolenic C18:3	2.300 ^c^	2.190 ^c^	2.030 ^b^	1.910 ^b^	1.827 ^b^	1.730 ^b^	1.527 ^a^	1.420 ^a^	2.307 ^c^	2.200 ^c^
Arachidic C20:0	0.586 ^a^	0.6017 ^a^	0.6300 ^a^	0.6910 ^b^	0.8767 ^b^	0.9610 ^c^	0.5550 ^a^	0.5717 ^a^	0.5350 ^a^	0.5713 ^a^
Eicosenoic C20:1	1.097 ^b^	0.992 ^c^	1.057 ^a^	0.953 ^ab^	1.037 ^a^	0.932 ^a^	1.090 ^b^	0.986 ^bc^	1.027 ^a^	0.922 ^a^
TSFA%	22.95 ^a^	21.51 ^a^	23.16 ^a^	21.61 ^a^	23.26 ^a^	21.70 ^a^	25.28 ^b^	23.52 ^b^	23.49 ^a^	21.93 ^a^
TUSFA%	80.78 ^c^	78.47 ^c^	80.67 ^c^	78.41 ^c^	80.55 ^c^	78.30 ^bc^	79.70 ^a^	77.48 ^a^	80.22 ^b^	78.00 ^b^

Different small letters in the same row indicate that the data are significantly different at *p* < 0.05. TSFA = total saturated fatty acids, TUSFA = total unsaturated fatty acids.

**Table 8 genes-13-00509-t008:** Estimates of variability parameters for the grain yield, grain shape (L/B ratio), and amylose content percentage in rice lines over two years.

Traits	GCV		PCV		GCV%		PCV%		GA		GA%	
	2019	2020	2019	2020	2019	2020	2019	2020	2019	2020	2019	2020
Grain yield (t/h)	1.88	2.76	2.49	3.15	11.82	13.51	13.63	14.45	2.45	3.2	21.12	26.04
Grain shape	0.008	0.024	0.011	0.029	3.36	5.62	3.8	6.1	0.17	0.3	6.12	10.68
A.C%	28.54	29.39	29.76	30.61	20.38	21.4	20.81	21.85	10.78	10.94	41.12	43.2

**Table 9 genes-13-00509-t009:** Estimates of variability parameters for chemical composition (%) of milled rice of the studied genotypes during the 2019 and 2020 growing seasons.

Chemical Composition	GCV		PCV		GCV%		PCV%		GA		GA%	
	2019	2020	2019	2020	2019	2020	2019	2020	2019	2020	2019	2020
Moisture	0.282	0.272	0.337	0.33	4.45	4.33	4.86	5.19	1	0.9	8.39	7.45
Crude Protein	0.08	0.07	0.105	0.098	3.62	3.43	4.137	4.044	0.51	0.47	6.53	6
Ether Extract	0.01	0.015	0.02	0.015	12.25	14.33	14.95	15.01	0.18	0.23	20.67	28.18
Ash	0.006	0.006	0.01	0.01	8.72	11.15	9.762	12.482	0.14	0.14	16.04	20.51
Available Carbohydrates	0.24	0.25	0.35	0.31	0.542	0.548	0.653	0.616	0.84	0.91	0.93	1.01

**Table 10 genes-13-00509-t010:** Estimates of variability parameters for gross chemical composition (%) and minerals content (mg/100 g) of stabilized rice bran samples of the studied genotypes during the 2019 and 2020 growing seasons.

Chemical Composition and Minerals Content	GCV		PCV		GCV%		PCV%		GA		GA%	
	2019	2020	2019	2020	2019	2020	2019	2020	2019	2020	2019	2020
Moisture	0.379	0.37	0.429	0.42	7.53	7.47	8.01	7.97	1.19	1.17	14.58	14.43
Crude Protein	0.36	0.33	0.39	0.37	3.51	3.41	3.67	3.58	1.18	1.13	6.92	6.67
Ether Extract	0.48	0.48	0.5	0.501	3.065	3.069	3.126	3.136	1.4	1.396	6.191	6.189
Ash	0.025	0.027	0.037	0.036	1.75	1.85	2.133	2.109	0.265	0.297	2.95	3.33
Crude Fiber	0.111	0.107	0.147	0.153	2.212	2.173	2.548	2.608	0.6	0.56	3.95	3.73
Available Carbohydrate	0.87	0.89	0.99	1	2.556	2.585	2.716	2.734	1.81	1.84	4.95	5.03
Phosphorus (P)	3583.3	3583.3	3693.3	3693.33	6.355	6.42	6.451	6.52	121.46	121.46	12.89	13.03
Potassium (K)	3572.3	3299.7	3662.3	3389.7	6.977	6.768	7.065	6.859	121.6	116.75	14.2	13.75
Magnesium (Mg)	92.035	92.035	95.335	95.335	7.022	7.127	7.147	7.25	19.418	19.418	14.21	14.43
Calcium (Ca)	10.242	10.366	11.79	11.93	9.4	10.09	10.08	10.827	6.144	6.181	18.04	19.37
Sodium (Na)	0.4	0.72	0.43	0.75	8.99	12.97	9.26	13.22	1.27	1.72	17.98	26.21
Manganese (Mn)	0.08	0.07	0.09	0.08	5.445	5.53	5.717	5.957	0.55	0.51	10.69	10.58
Zinc (Zn)	0.181	0.064	0.19	0.07	11.48	7.67	11.85	8.27	0.85	0.48	22.89	14.64
Iron (Fe)	0.526	0.521	0.548	0.542	9.724	9.9245	9.9301	10.122	1.463	1.457	19.62	20.05
Copper (Cu)	0.0089	0.005	0.0107	0.007	10.9	10.25	11.96	11.83	0.18	0.13	20.47	18.29

**Table 11 genes-13-00509-t011:** Estimates of variability parameters for some physical, chemical properties and fatty acids profile of crude rice bran oil from some rice genotypes (dry weight basis) of the studied genotypes during the 2019 and 2020 growing seasons.

Properties	GCV		PCV		GCV%		PCV%		GA		GA%	
	2019	2020	2019	2020	2019	2020	2019	2020	2019	2020	2019	2020
Refractive index (25 °C)	0.0001	0.0001	0.0001	0.0001	0.507	0.54	0.714	0.753	0.011	0.012	0.74	0.8
Specific gravity (25 °C)	0.00005	0.00005	0.00005	0.00005	0.7339	0.7324	0.7423	0.7426	0.0138	0.0137	1.49	1.49
Acid value (%)	0.134	0.136	0.139	0.136	15.08	15.32	15.366	15.33	0.74	0.76	30.49	31.53
Peroxide value (meq/kg oil)	0.1068	0.1072	0.107	0.108	20.171	20.299	20.221	20.345	0.672	0.673	41.45	41.72
Iodin value (gI/100 g oil)	7.752	7.752	7.759	7.759	2.5239	2.5243	2.525	2.5253	5.733	5.733	5.197	5.198
Saponification value (mg KOH/g oil)	7.66	7.45	8.23	7.85	1.494	1.477	1.549	1.5167	5.5	5.48	2.97	2.96
Unsaponifiable matter (%)	0.038	0.0384	0.0535	0.0527	5.2865	5.4879	6.2732	6.4296	0.3383	0.3444	9.18	9.65
Myristic C14:0	0.009	0.01	0.01	0.01	15.57	20.57	16.15	21.63	0.19	0.2	30.93	40.29
Palmitic C16:0	0.17	0.14	0.21	0.16	2.09	1.98	2.31	2.09	0.77	0.73	3.88	3.86
Palmitoleic C16:1	0.01	0.005	0.01	0.005	15.02	16.73	15.32	17.27	0.16	0.14	30.36	33.41
Stearic C18:0	0.029	0.045	0.04	0.05	8.39	11.66	9.619	11.751	0.303	0.434	15.06	23.82
Oleic C18:1	0.43	0.51	0.63	0.53	1.55	1.76	1.89	1.79	1.1	1.45	2.63	3.55
Linoleic C18:2	0.65	0.57	0.9	0.59	2.307	2.238	2.702	2.276	1.42	1.53	4.06	4.53
Linolenic C18:3	0.105	0.093	0.12	0.1	16.19	16.64	17.33	16.85	0.62	0.62	31.18	33.84
Arachidic C20:0	0.0272	0.02	0.0275	0.02	21.002	21.559	21.131	22.314	0.337	0.273	43	42.91
Eicosenoic C20:1	0.0009	0.001	0.0012	0.001	2.78	3.09	3.24	3.65	0.05	0.05	4.92	5.39
TSFA%	0.79	0.22	1.09	0.23	3.77	2.157	4.42	2.22	1.56	0.94	6.61	4.32
TUSFA%	0.18	0.16	0.21	0.18	0.529	0.506	0.569	0.549	0.81	0.75	1.01	0.96

**Table 12 genes-13-00509-t012:** Estimates of the percentage of advantage over commercial variety for the grain yield, grain shape (L/B ratio), and amylose content percentage of studied genotypes.

Traits Genotypes	Grain Yield (ton/h)		Grain Shape		A.C%	
	2019	2020	2019	2020	2019	2020
NRL 63	36.9 **	45.2 **	−7.8 **	−9.9 **	58.8 **	63.7 **
NRL 64	14.6 ns	23.6 **	−6.6 **	−8.7 **	61.5 **	67.3 **
NRL 65	16.9 *	23.2 **	−6.8 **	−9.8 **	66.9 **	71.0 **
NRL 66	33.2 **	38.7 **	−7.9 **	−14.5 **	23.1 **	25.4 **
L.S.D 5%	1.5	1.2	0.1	0.1	2.08	2.08
L.S.D 1%	2.2	1.7	0.1	0.2	3.02	3.03

L.S.D: least significant difference; ** highly significant at 1%; * significant at 5%; ns: non-significant.

**Table 13 genes-13-00509-t013:** Estimates of the percentage of advantage over commercial variety for chemical composition (%) of milled rice grains of the studied genotypes during the 2019 and 2020 growing seasons.

Traits Genotypes	Moisture		Crude Protein		Ether Extract		Ash		Total Carbohydrate	
	2019	2020	2019	2020	2019	2020	2019	2020	2019	2020
NRL 63	−4.8 ns	−4.8 **	2.4 ns	1.6 ns	41.2 **	47.0 **	14.8 **	19.4 **	−0.9*	−1.0 **
NRL 64	−7.6 **	−7.4 **	10.2 **	9.4 **	33.2 **	31.8 **	21.0 **	27.4 **	−1.6 **	−1.5 **
NRL 65	−1.4 ns	−1.6 ns	7.1 **	6.7 **	19.0 ns	19.7 **	3.7 ns	4.8 ns	−1.1 **	−1.1 **
NRL 66	−10.7 **	−10.9 **	4.0 *	3.6 ns	15.2 ns	7.6 *	−1.2 ns	−1.6 ns	−0.7 ns	−0.7 *
L.S.D 5%	0.6	0.4	0.3	0.3	0.1	0.05	0.07	0.07	0.6	0.5
L.S.D 1%	0.9	0.5	0.4	0.5	0.2	0.07	0.1	0.1	0.9	0.7

L.S.D: least significant difference; ** highly significant at 1%; * significant at 5%; ns: non-significant.

**Table 14 genes-13-00509-t014:** Estimates of the percentage of advantage over commercial variety for gross chemical composition (%) and minerals content (mg/100 g) of stabilized rice bran samples of the studied genotypes during the 2019 and 2020 growing seasons.

**Traits** **Genotypes**	**Moisture**		**Crude Protein**		**Ether Extract**		**Ash**		**Crude Fiber**	
	**2019**	**2020**	**2019**	**2020**	**2019**	**2020**	**2019**	**2020**	**2019**	**2020**
NRL 63	−13.8 **	−13.3 **	2.6 *	2.4 *	6.2 **	6.3 **	5.1 **	5.1 **	−1.3 ns	−1.1 ns
NRL 64	−0.2 ns	0.2 ns	−3.4 **	−3.6 **	8.2 **	8.2 **	1.7 ns	1.7 ns	−5.4 **	−5.3 **
NRL 65	4.6 ns	5.0 ns	−2.3 *	−2.3 *	7.5 **	7.6 **	1.1 ns	0.8 ns	−2.5 *	−2.5 ns
NRL 66	−8.6 **	−8.3 **	5.3 **	5.0 **	5.0 **	5.1 **	2.6 *	2.5 *	0.3 ns	0.3 ns
L.S.D 5%	0.42	0.42	0.34	0.35	0.26	0.27	0.21	0.17	0.4	0.4
L.S.D 1%	0.61	0.62	0.49	0.52	0.38	0.4	0.3	0.25	0.5	0.6
**Genotypes**	**Available Carbohydrates**		**Phosphorus (P)**		**Potassium (K)**		**Magnesium (Mg)**		**Calcium (Ca)**	
	**2019**	**2020**	**2019**	**2020**	**2019**	**2020**	**2019**	**2020**	**2019**	**2020**
NRL 63	−5.3 **	−5.4 **	1.0 ns	1.0 ns	−12.5 **	−12.1 **	1.5 ns	1.5 ns	12.0 **	12.68 **
NRL 64	−1.3 ns	−1.3 ns	−3.0 **	−3.1 **	−6.7 **	−6.5 **	−9.0 **	−9.1 **	−14.5 **	−15.59 *
NRL 65	−2.5 **	−2.4 **	−9.1 **	−9.2 **	−14.5 **	−14.1 **	6.7 **	6.8 **	−2.8 ns	−3.01 ns
NRL 66	−5.9 **	−5.9 **	−13.1 **	−13.3 **	−1.2 ns	−1.0 ns	9.8 **	9.9 **	2.9 ns	3.43 ns
L.S.D 5%	0.63	0.61	19.7	19.7	17.9	17.9	3.4	3.4	2.34	2.36
L.S.D 1%	0.92	0.89	28.7	28.7	26	26	5	5	3.41	3.43
**Genotypes**	**Sodium (Na)**		**Manganese (Mn)**		**Zinc (Zn)**		**Iron (Fe)**		**Copper (Cu)**	
	**2019**	**2020**	**2019**	**2020**	**2019**	**2020**	**2019**	**2020**	**2019**	**2020**
NRL 63	18.5 **	19.9 **	−10.3 **	−9.9 **	6.5 *	3.2 ns	1.6 ns	1.8 ns	−16.3 **	−16.7 **
NRL 64	−3.1 ns	−1.6 ns	2.1 ns	3.2 ns	32.4 **	20.0 **	−1.7 ns	−2.3 ns	−2.3 ns	−1.4 ns
NRL 65	10.9 **	24.7 **	−8.2 **	−7.9 **	25.9 **	16.6 **	11.0 **	10.9 **	7.0 ns	5.6 ns
NRL 66	17.4 **	31.8 **	−2.1 ns	−1.8 ns	13.1 **	7.2 **	23.4 **	23.7 **	14.0 **	12.5 *
L.S.D 5%	0.29	0.32	0.17	0.2	0.21	0.19	0.28	0.27	0.08	0.08
L.S.D 1%	0.43	0.46	0.25	0.29	0.3	0.28	0.41	0.4	0.12	0.12

L.S.D: least significant difference; ** highly significant at 1%; * significant at 5%; ns: non-significant.

**Table 15 genes-13-00509-t015:** Estimates of the percentage of advantage over commercial variety for some physical, chemical properties, and fatty acids profile of stabilized rice bran samples of the studied genotypes during the 2019 and 2020 growing seasons.

**Traits** **Genotypes**	**Refractive Index (25 °C)**		**Specific Gravity** **(25 °C)**		**Acid Value (%)**		**Peroxide Value (meq/kg Oil)**		**Iodin Value** **(gI/100 g Oil)**		**Saponification Value** **mg KOH/g Oil)**	
	**2019**	**2020**	**2019**	**2020**	**2019**	**2020**	**2019**	**2020**	**2019**	**2020**	**2019**	**2020**
NRL 63	−0.2 ns	−0.3 ns	−0.12 ns	−0.10 ns	26.0 **	26.2 **	50.9 **	50.9 **	1.0 **	1.0 **	3.3 **	3.3 **
NRL 64	0.5 ns	0.5 ns	0.11 ns	0.11 ns	16.5 **	16.3 **	64.4 **	65.6 **	1.8 **	1.8 **	2.9 **	2.8 **
NRL 65	−0.8 ns	−0.8 ns	1.09 **	1.09 **	37.0 **	37.3 **	39.1 **	39.3 **	4.6 **	4.6 **	1.4 **	1.3 **
NRL 66	0.7 ns	0.7 ns	1.54 **	1.54 **	50.0 **	50.4 **	77.0 **	77.4 **	−2.4 **	−2.4 **	3.7 **	3.7 **
L.S.D 5%	0.01	0.01	0.002	0.002	0.13	0.03	0.04	0.04	0.15	0.15	1.4	1.2
L.S.D 1%	0.02	0.02	0.003	0.003	0.2	0.04	0.06	0.06	0.22	0.22	2.1	1.7
**Genotypes**	**Unsaponifiable matter (%)**		**Myristic C14:0**		**Palmitic C16:0**		**Palmitoleic C16:1**		**Stearic C18:0**		**Oleic C18:1**	
	**2019**	**2020**	**2019**	**2020**	**2019**	**2020**	**2019**	**2020**	**2019**	**2020**	**2019**	**2020**
NRL 63	−4.2 ns	−4.3 ns	−33.9 **	−31.3 **	−0.2 ns	−0.1 ns	11.9 ns	15.2 **	5.5 ns	6.0 ns	−3.0 **	−3.0 **
NRL 64	8.1 *	8.4 *	−28.8 **	−16.5 **	−3.4 **	−2.5 **	−24.6 **	−15.2 **	−13.8 **	−11.4 *	−1.7 **	−0.7 ns
NRL 65	9.8 **	10.1 **	0.0 ns	0.5 ns	−4.4 **	−4.3 **	23.8 **	28.3 **	2.2 ns	1.3 ns	−0.1 ns	−0.1 ns
NRL 66	1.6 ns	1.6 ns	6.8 ns	8.2 *	−0.3 ns	−0.1 ns	16.7 *	19.6 **	16.0 **	17.2 **	1.6 **	1.6 ns
L.S.D 5%	0.23	0.23	0.09	0.04	0.28	0.33	0.05	0.03	0.13	0.15	0.43	0.83
L.S.D 1%	0.34	0.33	0.13	0.05	0.4	0.48	0.08	0.04	0.19	0.22	0.62	1.21
**Genotypes**	**Linoleic C18:2**		**Linolenic C18:3**		**Arachidic C20:0**		**Eicosenoic C20:1**		**TSFA%**		**TUSFA%**	
	**2019**	**2020**	**2019**	**2020**	**2019**	**2020**	**2019**	**2020**	**2019**	**2020**	**2019**	**2020**
NRL 63	4.2 **	4.4 **	−0.9 ns	−1.1 ns	5.3 ns	5.3 ns	6.8 **	7.6 **	−2.3 ns	0.6 **	0.7 **	0.6 **
NRL 64	2.0 **	2.5 ns	−16.5 **	−13.7 **	11.7 ns	20.9 **	2.9 ns	3.3 ns	−1.4 ns	0.5 *	0.6 *	0.5 **
NRL 65	1.6 *	1.7 ns	−21.4 **	−21.2 **	68.4 **	68.2 **	1.0 ns	1.1 ns	−1.0 ns	0.4 ns	0.4 *	0.4 ns
NRL 66	−2.0 **	−2.0 ns	−35.5 **	−34.6 **	0.0 ns	0.1 ns	6.2 **	7.0 **	7.6 **	−0.7 **	−0.6 **	−0.7 **
L.S.D 5%	0.43	0.85	0.1	0.2	0.08	0.04	0.03	0.04	1.03	0.32	0.32	0.32
L.S.D 1%	0.62	1.23	0.14	0.29	0.12	0.06	0.05	0.05	1.5	0.46	0.46	0.46

TSFA = total saturated fatty acids, TUSFA = total unsaturated fatty acids; L.S.D: least significant difference; ** highly significant at 1%; * significant at 5%; ns: non-significant.

## Data Availability

Relevant data applicable to this research are within the paper.

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
