# Peer review of "Assessment of Genetic Variability and Bran Oil Characters of New Developed Restorer Lines of Rice (Oryza sativa L.)"

_genes, 2022, doi:10.3390/genes13030509_

Round 1
Reviewer 1 Report
1, Please read through this whole manuscript to check the grammar mistake.
For example, line 50/51, there is a half sentence of "While the production of rice in Egypt was 6.00 million tons [1]. ".
Then in line 52/53, there is another half sentence "A staple diet for more than two billion people because it contains minerals, vitamins and fiber, and carbohydrates [2, 3]".
In line 54/55, it was written as "Rice bran is an inexpensive by-product of raw rice [4]. Rice bran is a major by-product of the rice milling process accounting for 5–10% of milled rice. " These two sentences are repeating the same information. Please rewrite it.
2, I would suggest the authors reorganize the introduction section from line 48 to line 92. Too many sentences started with Rice bran. There is a less of smooth/logic transition.
3, All data were presented as tables. I would suggest the authors could present the data as bar graphs including all standard deviations, which could make the data much easier to understand.
Author Response
The attached file the response.

Reviewer 2 Report
Manuscript Number: GENE-1593447 titled “Assessment of Genetic Variability and Bran Oil Characters of New Developed Lines of Rice (Oryza sativa L.)” focused on the characterization of the chemical component of rice bran and bran oil composition in new developed rice lines. Rice is one the most important staple food feeding almost half of the world population. Based on the literature review provided by the authors, rice bran content a wide range of important products as the bran oil does. On this basis, characterizing the rice bran and bran oil composition of the new rice developed varieties is quiet relevant as evidenced by the significant amount of data recorded in this manuscript. The authors noticed high phenotypic and genotypic coefficients of variation of amylose content (%), peroxide value (meq/kg oil), myristic C14:0, and arachidic C20:0 which suggests an environment effect. The authors ended this paper by hilighting the fact that the high genetic advance as a percentage of mean recorded in most traits suggested that heritability was due to additive gene effects. Despite these significant acheivements, the paper needs deep modifications.
Specific comments
- Introduction may describe the importance of the crop used, the problematic and the objectives of the topic.
- Line 89, it looks like that the sentence is incomplete.
- The methodologies used to analyze the chemical products have to be clearly described. In the other words the methodological section needs deep modifications.
- For others comments please check the attached pdf file.

Author Response
The manuscript entitled in " Assessment of Genetic Variability and Bran Oil Characters of New Developed Lines of Rice (Oryza sativa L)" is very well written and suitable for publication in Coatings. However, some revisions below are necessary.
Are the methods adequately described?
Yes, the methods are adequately described : 157-218
Line 125-127 is changed
2.1. Plant materials
Five genotypes of rice include four newly developed restorer lines namely NRL63, NRL64, NRL65, NRL66, and Giza 178 as check variety. These lines were selected of
Are the results clearly presented?
Results
Discussion
Line 601-603 : is deleted
In general, the phenotypic coefficient of variation was higher than the genotypic coefficient of variation suggesting an influence of environment on the expression of these characters
Are the conclusions supported by the results?
Conclusion is modified :
In conclusion, analysis of variance showed there are significant differences among the genotypes for all the characters under study. This indicated that there is scope for the selection of promising genotypes from the present set of genotypes for yield and other traits improvement. The magnitude of the phenotypic coefficient of variation was higher than the genotypic coefficient of variation for all the studied characters. Both GCV and PCV were high for amylose content (%), peroxide value (meq/kg oil), arachidic C20:0 of stabilized rice bran oil. While both GCV and PCV were moderate for grain yield (t/h), ether extract, ash content, of milled rice, calcium, sodium, zinc, copper, acid value (%), myristic C14:0, palmitoleic C16:1, stearic C18:0 and linolenic C18:3 of stabilized rice bran oil. The genetic advance in the percentage of mean was high for grain yield (t/h), amylose content (%), ether extract and ash of milled rice, sodium, zinc, iron, and copper of stabilized rice bran oil, acid value (%), peroxide value (meq/kg oil) and myristic C14:0, palmitoleic C16:1, stearic C18:0, linolenic C18:3, arachidic C20:0 of stabilized rice bran oil. Moreover, the genetic advance in the percentage of mean was moderate for grain shape, moisture, phosphorus, potassium, magnesium, calcium, magnesium, stearic C18:0, of stabilized rice bran oil. The percentage of advantage over Giza 178 commercial variety was significant and highly significant among the genotypes for most of the studied characters in the two years, indicating that the selection is effective in the genetic improvements for these traits. The lines NRL 66 and NRL 64 showed an advantage over Giza 178 commercial variety from 5.1 to 8.2 % for ether extract of stabilized rice bran, while, the lines NRL 63 and NRL 66, showed an advantage values 36.9 and 33.2 % for grain yield (t/h), at the first and second year, respectively.
Specific comments :
- Introduction may describe the importance of the crop used, the problematic and the objectives of the topic.
It is improved
- Line 89, it looks like that the sentence is incomplete.
It is changed :
This healthy oil is also a rich source of monounsaturated fatty acids (n-9 MUFA), n-6 PUFA and sterols, as well as it has been shown to reduce bad cholesterol.
- The methodologies used to analyze the chemical products have to be clearly described. In the other words the methodological section needs deep modifications.
The methods are adequately described : 157-218
2.3. Amylose content:
The amylose content of the native rice starch was determined according to the method described for the analysis of milled rice amylose content by [23].
2.4. Processing of rice bran:
Different substances like husk may be present in the bran. Hence, the full fatted raw bran was sieved which removes husk. The samples thus obtained were free from impurities.
2.5. Stabilization of rice bran:
Rice bran samples were stabilized by autoclave under atmospheric pressure for 10 min. at 120°C according to the method described by [24]. Finally, bran samples were stored in dark conditions at − 10°C in water insusceptible containers until further analyses.
2.6. Extraction of rice bran oils:
Rice bran oil was extracted according to the method described by [25].
2.7. Rice bran analysis:
2.7.1. Determination of gross chemical composition:
Moisture, ether extract, crude protein (N × 5.95), ash, and crude fiber contents were performed according to the Association of Official Analytical Chemists [26]. Available carbohydrates were determined by difference according to the methods of [26]. Minerals contents (Ca, Mg, Fe, Cu, Zn, and Mn) were determined according to the methods outlined in [26] using atomic absorption spectrophotometer (Perkin Elmer Model 4100 ZL), while (Na and K) were determined using a flame photometer. On the other hand, phosphorus was determined by the ascorbic acid technique using the colorimetric method.
2.7.2. Determination of Fatty acids composition of rice bran oil samples:
The methyl esters were prepared using benzene: methanol: concentrated sulfuric acid (10:86:4) and the methylation process was carried out for one hour at 80-90°C according to [27]. Identification of the fatty acid methyl esters was performed by G.L.C A pyeunic'am gas-liquid chromatography (model 4550) equipped with a flame ionization detector and coiled glass Colum (1.6 m × 4 mm) packed with 10% PEGA (polyethylene glycol adipate) supported on chromosorb W-AW 100-200 mesh. Samples (1-1.5 ul) into the column using ahamilton microsyringe. Gas chromatographic conditions used for isothermal analysis were column 190°C Flow rates: Hydrogen 33 ml/min. nitrogen 30 ml/min. and air 330 ml/min. Peak areas were measured using spectto physic integrator, [26].
- For others comments please check the attached pdf file.
In abstract : it is modified :
Abstract: Rice is the most important crop in Egypt. In view of the existence of a gap between the demand and availability of local edible oils and in order to raise the nutritional value of rice and thus improve the nutritional value of the consumer. This investigation was carried out at the Experimental Farm of Sakha Agricultural Research Station, Sakha, Kafr El-Sheikh, Egypt, during the 2018 and 2019 seasons, using five newly developed genotypes of rice, namely NRL63, NRL64, NRL65, NRL66, and Giza 178 as check variety to evaluate the analytical characterization of raw rice bran and rice bran oil from rice bran, study the genetic variability and genetic advance for various quantitative and qualitative traits in rice as well as, rice bran oil. The genotypes were evaluated in a randomized complete block design (RCBD) with three replications. Analysis of variance revealed highly significant variations among the genotypes for all the studied characters. Data revealed that high estimates of the phenotypic coefficient of variance (PCV%) and genotypic coefficient of variance (GCV%) were observed for amylose content (%), peroxide value (meq/kg oil), myristic C14:0, and arachidic C20:0, indicating that they all interacted with the environment to some extent. The line NRL66 and NRL64 showed the highest and high values of mean performance for grain yield (t/h), grain type (shape), amylose content (%), crude protein, ether extract and ash of milled rice, crude protein, ether extract, ash, phosphorus, magnesium, manganese, zinc, and iron of stabilized rice bran oil. Genetic advance as a percentage of mean was high for most of the studied traits. It indicates that most likely the heritability is due to additive gene effects and selection may be effective. The percentage of advantage over the Giza 178 as commercial variety was significant and highly significant among the genotypes for all the characters studied in the two years, indicating that the selection is effective in the genetic improvements for these traits.
Introduction :
Rice (Oryza sativa L) is
Line 52-53 were deleted
Materials and Methods
Line 123-124 was corrected
2.1. Plant materials
Five genotypes of rice include four newly developed restorer lines namely NRL63, NRL64, NRL65, NRL66, and Giza 178 as check variety. These lines were selected of a set
- The length of the row was 5 m in length and 20 cm between rows, each row has 25 individual plants.
- (gI/100 g oil),
- ml/min. and air 330 ml/min. Peak areas were measured using spectto physic integrator, [26].
σ2g = (MSS-MSE)/r
- Where : MSS : Mean sum of squares due to treatments, MSe : Mean sum of squares due to error from the analysis of variance, and r : Number of replications
- Results
- This paragraph is removed
- A set of 122 iso-cytoplasmic restorer lines of rice were developed from two promising rice hybrids in Egypt IR79156A/86945-L and G46A/Giza 178. Three iso-cytoplasmic restorer lines were identified and selected from the hybrid IR79156A/86945-L, moreover, one line was selected from the hybrid G46A/Giza 178. Evaluated four newly developed lines and Giza 178 as local check variety for grain yield, some grain quality traits, and bran oil characters.
- The average performance of all the iso-cytoplasmic restorer lines derived from two hybrids was observed. Iso-cytoplasmic restorer lines with better performance over other lines were also given due importance.
- Table 2. Mean performances for rice grain yield, grain shape (L/B ratio), and amylose content percentage traits of the studied genotypes during the 2019 and 2020 growing season.
- The comma (,) has been added in all required positions :
- season,
- The distance has been added in all required positions in mg/100 g
- The font size has been modified and standardized in all required positions in 2019 and 2020
- Table 3. Mean performances for Chemical composition (%) of milled rice grains of the studied genotypes during the 2019 and 2020 growing season.
- Table 5. Mean performances for minerals content (mg / 100 g) of stabilized rice bran samples of the studied genotypes during the 2019 and 2020 growing season.
- The highest magnesium content was observed in NRL 66 rice bran 147.29 mg/100 g, in the first year. The levels of calcium in the bran ranged from 29.27 to 38.31 mg/100 g rice bran sample at the first year. The highest calcium content was observed in rice bran of NRL 63, 38.31 mg/100 g, in the first year. Furthermore, the iron levels varied within a range of 6.86 – 8.61 mg/100 g rice bran samples at the first year. Rice bran of NRL 66 was higher iron content than other rice bran genotypes. Apparent also from the same Table that, stabilized rice bran of NRL 66 had the highest elements content in comparison with the other tested genotypes. Concerning the levels of Sodium in the bran ranged from 5.61 to 7.70 mg/100 g rice bran sample at the first year. The highest Sodium content was observed in rice bran of NRL 63 and NRL 66, with the values 7.70 and 7.63 mg/100 g, in the first year. Furthermore, the Manganese levels varied within a range of 4.51 – 5.45 mg/100 g rice bran samples at the first year. Rice bran of NRL 64 was higher Manganese content than other rice bran genotypes. Regarding the levels of Zinc was ranged from 3.21 to 4.25 and 3.01 to 3.61 mg/100 g samples at the first and second years, respectively. The highest Zinc content was observed in rice bran of NRL 64 and NRL 65, with the values 4.25 and 4.04 mg/100 g, in the first year.
- Table 8: Analysis of variance for the grain yield, grain shape (L/B ratio), and amylose content percentage of the studied genotypes during 2019 and 2020 growing season.
- Table 9. Analysis of variance for chemical composition percentage of milled rice grains of the studied genotypes during the 2019 and 2020 growing season.
- Table 10. Analysis of variance for gross chemical composition (%) and minerals content (mg /100 g) of stabilized rice bran samples of the studied genotypes during the 2019 and 2020 growing season.
- Table 11. Analysis of variance for some physical, chemical properties, and fatty acids profile of rice bran oil from some rice genotypes (dry weight basis) of the studied genotypes during the 2019 and 2020 growing season.
- S.D. : The least significant difference
Round 2
Reviewer 1 Report
It has been much better improved.
Author Response
The reviewer comment: It has been much better improved.
Reviewer 2 Report
All the suggestions required have been followed according the recommendations of the reviewers, this paper can be accepted for publication.
Author Response
The reviewer comment: All the suggestions required have been followed according the recommendations of the reviewers, this paper can be accepted for publication.